# Associations between gestational weight gain under different guidelines and adverse birth outcomes: A secondary analysis of a randomized controlled trial in rural western China

Yingze Zhu[1], Liang Wang[1], Qi Qi[1], Yue Cheng[2], Zhonghai Zhu[1,3,4]*, Lingxia Zeng[1,3,4]*

1 Department of Epidemiology and Biostatistics, School of Public Health, Xi'an Jiaotong University Health Science Center, Xi'an, Shaanxi, P.R. China, 2 Department of Nutrition and Food Safety Research, School of Public Health, Xi'an Jiaotong University Health Science Center, Xi'an, Shaanxi, P.R. China, 3 Center for Chronic Disease Control and Prevention, Global Health Institution, Xi'an Jiaotong University, Xi'an, China, 4 Key Laboratory for Disease Prevention and Control and Health Promotion of Shaanxi Province, Xi'an, China

* tjzlx@mail.xjtu.edu.cn (LZ); zhuzhonghai@hotmail.com (ZZ)

**Data Availability Statement:** All data can be found in the manuscript and supporting information files.

## Abstract

Several gestational weight gain (GWG) guidelines have been established based on mono-center or multicenter researches. We aimed to examine the associations between categories of GWG under the Institute of Medicine (IOM) recommendation guideline, the Chinese National Health Commission (NHC) guideline, and weight-gain-for-gestational-age z-scores derived from the INTERGROWTH-21st Project and adverse birth outcomes. We used data from an antenatal micronutrient supplementation trial in rural western China between 2002 and 2006. Maternal weekly average GWG during the second and third trimesters was calculated and classified into inadequate, adequate and excessive GWG according to the IOM and NHC, respectively. Weight-gain-for-gestational-age z-scores derived from the INTERGROWTH-21st Project were grouped into three subgroups using two approaches: z-score percentile<25th, 25th to 75th, >75th and z-score <-1, -1 to 1, >1 SD. Infant birth weight and gestational age were measured using standard approaches. Generalized linear model with binomial family and logit link was applied to estimate the odds ratio (OR) and 95% confidential intervals (CI) for GWG categories and adverse birth outcomes. Among 1,239 women with normal weight (18.5 kg/m² to 23.9 kg/m²) during early pregnancy, 18.0% and 34.2% were classified as adequate GWG according to IOM and NHC, respectively. Less than half of Chinese women reached optimal GWG by any recommendation guideline. According to NHC, excessive GWG showed a significant association with macrosomia (OR 3.75, 95% CI 1.03, 13.74), large-for-gestation-age (LGA) (OR 2.12, 95% CI 1.01, 4.45), and inadequate GWG was associated with post-term birth (OR 2.25, 95% CI 1.21, 4.16), compared with adequate GWG. Inappropriate GWG was associated with adverse birth outcomes even among women with normal weight during early pregnancy. The monitoring and interventions of weight status during pregnancy, especially for the second and third trimesters, are of

**Funding:** This work was supported by the National Natural Science Foundation of China (grant 81872633 to LZ and 82103867 to ZZ), China Postdoctoral Science Foundation (grant 2021M702578 to ZZ) and National Key Research and Development Program of China (grants 2017YFC0907200 and 2017YFC0907201). The funders had no role in study design, data collection and analysis, decision to publish, or preparation of the manuscript.

**Competing interests:** The authors have declared that no competing interests exist.

great public health importance for optimal birth outcomes. Additionally, developing guideline of appropriate GWG ranges should account for the traits of regional population.

## Introduction

Adverse birth outcomes such as small-for-gestational-age (SGA) and preterm birth, lay a disadvantaged foundation for later development, with complex genetic and environmental causes [1, 2]. The prevalence of SGA in low- and middle-income countries (LMICs) was as high as 41.5% [3], and the highest prevalence of macrosomia in 2013 was 14.9% in Algeria [4]. In China, data from perinatal health care surveillance system in 12 cities showed that the rate of macrosomia rose from 6.00% in 1994 to 8.49% in 2000 [5]. Maternal weight status has been identified as a highly predictive factor of fetal growth, which could be considered a proxy of the intrauterine environment [6]. Accordingly, weight gain during pregnancy, which reflects the growth of the fetus, placenta, and uterus, is another independent predictor of fetal development [7]. Therefore, to ensure a healthy fetus, establishing recommendation range for gestational weight gain (GWG) is necessary. The United States Institute of Medicine (IOM) promulgated recommended ranges of GWG based on the lowest prevalence of common pregnancy outcomes in 2009, which has been the most widely adopted and commonly applied guideline in clinical practice [8]. An individual participant data (IPD) meta-analysis revealed that, compared with maternal GWG within the IOM recommendation ranges (defined as adequate GWG), inadequate GWG showed a 1.94 times higher risk of preterm and 1.52 times higher risk of SGA [9]. Additionally, for excessive GWG relative to adequate GWG, the risk of large-for-gestational-age (LGA) increased (adjusted odds ratio (aOR): 2.00; 95% confidence interval (CI) 1.58, 2.54) and the risk of SGA reduced (aOR: 0.66; 95% CI 0.50,0.87) [9].

However, whether the predetermined cutoffs in IOM guideline could minimize adverse maternal and neonatal outcomes among population residing in LMICs remains no consensus. A meta-analysis included studies from the USA and western Europe suggested that GWG above IOM guideline was associated with macrosomia (OR 1.87; 95% CI 1.70, 2.06) [10]. While, meta-analysis conducted among sub-Saharan Africa researches found no significantly association between excessive GWG and macrosomia after applying IOM guidelines [11]. The establishment of IOM was based on USA white women and mainly examined in developed settings, where the incidence of overweight and obesity is much higher than those women residing in LMICs [10]. Therefore, considering the differences in ethnicities, dietary patterns, environmental exposure and other related factors, many countries have built their cohort to explore the optimal GWG ranges for local populations in recent years [12–14].

In 2022, the National Health Commission (NHC) of the People's Republic of China issued a *Standard of Recommended for weight gain during pregnancy period* and implemented on October 1st, 2022 [15]. This recommendation utilized data from nine cities in China (including Beijing, Wuhan, Chengdu, Shenzhen, Dongguan, Harbin, Shijiazhuang, Qingdao and Danyang) and considered composite maternal and neonatal outcomes. In addition, other methods to define optimal GWG e.g., z-scores are developed [13, 16]. The Fetal Growth Longitudinal Study of the INTERGROWTH-21st Project conducted in eight geographically diverse urban regions in Brazil (Pelotas), China (Beijing), India (Nagpur), Italy (Turin), Kenya (Nariobi), Oman (Muscat), the United Kingdom (Oxford), and the United States (Seattle), described patterns of GWG among healthy, well-nourished, normal weight (BMI of 18.5–24.9 kg/m$^2$) and educated women that are compatible with desirable healthy pregnancy outcomes, and then generated weight-gain-for-gestational age z-score standards [17]. However, the cut-

off values of this GWG z-score guiding clinical practice have not been identified by robust epidemiological evidence across regions.

Studies mentioned above were conducted in relatively developed cities in China and did not include the population in rural western areas. Health conditions of women in reproductive age varied across different socioeconomic groups and geographies in China. For example, the prevalence of anemia in women of reproductive age was 45.7% in rural western areas of China in 2005, while corresponding average prevalence was 19.9% in 2002 in China according to China Health and Nutrition Survey [18]. In addition, none investigation has examined the appliance of weight-gain-for-gestational age z-score constructed by the INTERGROWTH-21st Project among Chinese women. Therefore, whether the existing three GWG guidelines above could applied to the population in rural western areas in China and their associations with adverse birth outcomes remain unknown.

In the present study, we used data from a cluster-randomized, double-blind trial in rural western China, where populations suffered from relatively disadvantaged socioeconomic status, with limited health workforce, health infrastructure, and maternal care [19]. We aimed to examine the associations between GWG classifications and adverse birth outcomes among rural western Chinese mother-infant pairs under the NHC and IOM guidelines, respectively. In addition, we aimed to explore the associations between the weight-gain-for-gestational-age z-scores constructed by the INTERGROWTH-21st Project and birth outcomes. We hypothesized that the GWG recommend ranges released by NHC would be more suitable for Chinese women in rural western China. In addition, these results would help to expand the generalization of appropriate GWG guidelines for optimizing birth outcomes in other rural areas in LMICs.

## Materials and methods

### Study design and participants

We used secondary data from a cluster-randomized, double-blind trial in rural western China between August 2002 and February 2006 (ISRCTN08850194). This trial primarily aimed to investigate the effects of prenatal micronutrient supplementation on birthweight and neonatal mortality. The details of this trial have been described previously [20]. Briefly, all pregnant women less than 28 week's gestation from every village in two poor rural counties were randomized by village to take a daily capsule of folic acid, iron-folic acid (IFA), or multiple micronutrients (MMN) until delivery. Ethical approvals for the trial were received from the Ethics Committee of Xi'an Jiaotong University Health Science Center (No 2002001). Furthermore, written informed consent was obtained from mothers.

### Measurements

**Maternal body mass index and gestational weight gain.** Pregnant women enrolled in the prenatal trial could receive free antenatal care for at least three times, at which trained maternal and child health staff measured their height and weight using standardized equipment and procedures. Maternal BMI during early pregnancy (<14 weeks) was used as a proxy for pre-pregnancy BMI in the present study to classify GWG, given that maternal weight changes usually occur after the first trimester [21]. Therefore, we used data from women who enrolled before 14 weeks of gestation in the final analysis.

In addition, to focus on the GWG regardless of maternal malnutrition status (underweight and overweight/obesity) pre-pregnancy, data from pregnant women with normal weight were included in the final analysis. Due to the differences of normal weight using BMI cut-offs by IOM ($18.5 \text{ kg/m}^2 \leq \text{BMI} < 25.0 \text{ kg/m}^2$) and NHC ($18.5 \text{ kg/m}^2 \leq \text{BMI} < 24.0 \text{ kg/m}^2$), we restricted

the sample size to pregnant women with normal weight defined by the NHC, a narrower criterion.

We calculated weekly gestational weight gain (kg/week) during the second and third trimesters as a predictor because it is independent of gestational age. The recommended range by IOM was 0.35–0.50 kg/week for women with normal weight, while the NHC recommended 0.26–0.48 kg/week for women with normal weight (detailed cut-off values under different recommendations was shown in S1 Table). Inadequate, adequate and excessive GWG were defined as below, within, and above the recommended range, respectively.

According to the Fetal Growth Longitudinal Study of the INTERGROWTH-21st Project standards, we calculated weight-gain-for-gestational age z-score and z-score percentiles for each pregnant woman [17]. However, given the unknown cut off points of this z-score, we classified the z-score into three groups by general rules in two ways: category 1 based on percentiles of z-score (z-score percentile<25th, 25th to 75th, >75th) and category 2 based on origin z-score (-1, -1 to 1, >1 SD).

**Infant birth outcomes.** Birth weight was measured within one hour of delivery by hospital nursing staff using an electronic scale (type BD 585, Tanita, Dongguan, Guangdong Province, China) if newborns were delivered in the six counties hospital and three largest township hospitals. For other smaller township hospitals and home births, birth weight was measured with a baby scale (type RTZ-10A-RT, Wuxi Weigher Factory, Wuxi, China). Low birth weight (LBW) was defined as birthweight <2500 g, and macrosomia was defined as birthweight >4000 g. Gestational age at birth was measured as completed days based on the first day of the last menstrual period obtained at the baseline interview of enrollment. Preterm delivery was defined as delivery before 37 gestational weeks, and post-term birth was defined as delivery after 42 gestational weeks. SGA and LGA babies were defined as having a birth weight of less than the 10th or more than the 90th percentile for gestational age (INTERGROWTH-21st Project) [22].

**Covariables.** Covariables were collected by face-to-face interviews in the parent trial following standard procedures. Common confounders were selected based on the literatures including parental sociodemographic characteristics and maternal nutrition status. We considered the following confounding factors: parental age in years (continuous variable), education attainment level (did not complete primary school, primary school, secondary school, high school diploma or greater), occupation (farmer, others), household wealth (low, medium, high), maternal mid-upper arm circumference at the enrollment (continuous variable), parity (0, 1, ≥2), randomized regimens (folic acid, folic acid plus iron and multiple micronutrients), pre-pregnancy medical history, gestational week in early pregnancy when the maternal weight was measured (continuous variable) and offspring sex (male, female). The household wealth index at the enrollment was constructed using principal component analysis for household assets and dwelling characteristics and was further categorized by its terciles to defined as low-, medium- and high-household wealth. The most prevalent disease during pregnancy was anemia in our sample, with a prevalence of 19.13% among women with normal weight, while other disease had a very low prevalence, ranging between 0.08% and 0.40%. In our sample, only nine women suffered from two diseases (three women with cardiac disease and anemia, three with kidney disease and anemia, two women with chronic liver disease and anemia, one woman with hypertension and anemia). Only one woman suffered from chronic liver disease, hypertension, and hyperthyroidism. Therefore, due to the low frequency of other disease except for anemia, women have any one of these cardiac disease, kidney disease, chronic liver disease, hypertension, anemia or hyperthyroidism, was considered to have a preconception medical history for simplicity and convergence of statistical models.

**Statistical analysis.** 4,604 singleton births were obtained from the parental trial, while among them 2,252 after 14 weeks of gestation age were not available for the analysis. Of the remaining 2,352 participants, 688 women had missing values on weight data during the second or third trimester and 98 infants had missing values on birth weight or gestational age at birth. Among the remaining 1,566 participants, 327 women with underweight or overweight/obesity were excluded from main analysis for the different cut-off points of normal weight using BMI by IOM and NHC guidelines. Finally, 1,239 mother-infant pairs were included in the main analysis (as shown in S1 Fig).

Continuous variables were presented as mean and standard deviations (SD) and categorical variables were expressed as numbers (%, percentage). The Kappa test was used to compare GWG classifications by different approaches. Generalized linear model with binomial family and logit link was applied to examine associations between GWG categories and adverse birth outcomes (including preterm delivery, post-term delivery, LBW, macrosomia, SGA, and LGA), with adequate GWG or middle subgroup set as reference. Confounder mentioned above were adjusted in each regression model. The areas under the curve (AUC) for adverse birth outcomes predicted by each cut-off value were also calculated. In addition, birthweight, birthweight for gestational age z-score and gestational age at birth were treated as continuous outcomes fitted with a generalized linear model. The associations of continuous weekly average GWG during the second and third trimesters and weight-gain-for-gestational-age z-scores derived from INTERGROWTH-21[st] Project with birth outcomes (both dichotomous and continuous variables) were also examined.

In addition, we examined the associations of GWG classifications under IOM and NHC guidelines with adverse birth outcomes among available pregnant women with thinness, normal weight or overweight/obesity. To explore if the association was modified by infant sex or parental education level, the interaction term was added to the model to assess the statistical significance. For supplementary analyses, we compared the baseline characteristics between women included in the main analysis and those were not. Further, we conducted inverse probability weighting (IPW) to address the potential of selection bias due to the missing values. IPW was constructed using covariables mentioned above. Additionally, we applied E-value to assess the potential effect of unmeasured confounders [23].

All *P* values were 2-sided with an alpha of 0.05. Statistical analyses were conducted using Stata 15.0 (Stata Corp, College Station, Texas, USA).

## Results

A total of 1,239 mother-infant pairs were included in the final analysis (as shown in S1 Fig). Table 1 presents the baseline characteristics of participants, showing that the majority of parents had a secondary education level and were farmers. Among offspring, 55.9% were boys, and the prevalence of preterm, post-term birth, LBW, macrosomia, SGA, and LGA was 3.7%, 5.8%, 3.9%, 1.5%, 13.6%, and 4.9%, respectively.

Table 2 shows that most women did not achieve the recommended range for GWG defined by each classification. The proportion of women who achieved optimal GWG velocity during the second and third trimesters under IOM and NHC guideline was 18.0% and 34.2%, respectively. More women were classified as inadequate GWG based on IOM guideline compared with NHC guideline (64.4% versus 46.7%). Additionally, the Kappa value between IOM and NHC recommendation was 0.68. For categorical groups of INTERGROWTH-21[st] Project z-score, the proportion of the middle group was 28.4% (z-score percentiles in 25[th] to 75[th]) and 39.8% (z-score in -1 to 1 SD).

**Table 1. Characteristics of the pregnant women and their offspring (n = 1,239).**

| Factors | No. (%) |
|---|---|
| *Pregnant women characteristics* | |
| **Maternal age (years) /Mean (SD)** | 24.5 (4.5) |
| **Paternal age (years) /Mean (SD)** | 27.7 (4.1) |
| **Maternal education** | |
| < 3 years | 63 (5.1) |
| Primary | 301 (24.4) |
| Secondary | 670 (54.2) |
| High school and above | 202 (16.3) |
| **Paternal education** | |
| < 3 years | 9 (0.7) |
| Primary | 169 (13.7) |
| Secondary | 709 (57.4) |
| High school and above | 349 (28.2) |
| **Maternal occupation** | |
| Farmer | 1000 (81.2) |
| Others | 231 (18.8) |
| **Paternal occupation** | |
| Farmer | 872 (70.4) |
| Others | 366 (29.6) |
| **Parity at enrollment** | |
| 0 | 817 (65.9) |
| 1 | 378 (30.5) |
| ≥2 | 44 (3.5) |
| **Maternal MUAC (cm)** | |
| <21.5 | 171 (13.9) |
| ≥21.5 | 1062 (86.1) |
| **Randomized regimens** | |
| Folic acid | 456 (36.8) |
| Folic acid plus iron | 421 (34.0) |
| Multiple micronutrient | 362 (29.2) |
| **Household wealth at enrollment[a]** | |
| Low | 376 (30.4) |
| Medium | 418 (33.7) |
| High | 445 (35.9) |
| *Infant characteristics* | |
| **Sex** | |
| Male | 693 (55.9) |
| Female | 546 (44.1) |
| **Birth weight (gram) /Mean (SD)** | 3189 (411) |
| **Birthweight for gestational age z-score/Mean (SD)** | -0.2 (1.0) |
| **Gestational age at birth/Mean (SD)** | 39.8 (1.5) |
| **Preterm birth (< 37 gestational weeks)** | 46 (3.7) |
| **Post-term birth(≥42 gestational weeks)** | 72 (5.8) |
| **Low birth weight (< 2500 g)** | 48 (3.9) |
| **Macrosomia(>4000 g)** | 19 (1.5) |
| **Small-for-gestational age (<10th percentile)** | 169 (13.6) |

*(Continued)*

**Table 1.** (Continued)

| Factors | No. (%) |
|---|---|
| **Large-for-gestational age (>90th percentile)** | 61 (4.9) |

Abbreviations: SD: standard deviations; MUAC, mid–upper arm circumference.

[a]Household wealth at enrollment was derived from principal component analysis for household assets and dwelling characteristics, and was further categorized by its terciles.

Table 3 wshows the associations between different GWG classifications and adverse birth outcomes, and statistically significant associations were only observed for the NHC category. We found that compared to adequate GWG, inadequate GWG contributed to a 2.25 (95% CI 1.21, 4.16) times higher risk for post-term birth. In comparison, excessive GWG contributed to a 3.75 (95% CI 1.03, 13.74) times higher risk for macrosomia and a 2.12 (95% CI 1.01, 4.45) times higher risk for LGA compared with adequate GWG. However, we observed no significant association between GWG classifications and continuous birth outcomes, including birth weight, birthweight for gestational age z-score, and gestational age at birth (see S2 Table). As shown in Fig 1 and S3 Table, the AUC of different birth outcomes predicted by each cut-off value was between 0.6162 and 0.8058. Although the AUC values of adverse birth outcomes showed no statistically significant difference among different approaches, the point estimate size of NHC guideline is numerically higher than IOM guideline. The statistical power of test might be limited due to the sample size.

Continuous weekly average weight gains during the second and third trimesters and INTERGROWTH-21st Project z-score were also examined. We observed a positive association between z-score and birth weight (for per SD increase of GWG z-score, mean difference (MD) 21.57g; 95% CI 3.48, 39.67), birthweight for gestational age z-score (MD 0.05; 95% CI 0.01, 0.10), and LGA (OR 1.28; 95% CI 1.01, 1.61) (see S4 Table). However, no significant association was observed between z-score subgroups and birth outcomes (both continuous outcomes and dichotomous outcomes). In addition, by including all pregnant women with thinness, normal weight or overweight/obesity in the analyses, the classifications based on IOM or NHC were shown in S5 Table. The association between categorical predictors and adverse birth outcomes is shown in S6 Table. All these results showed similar associations presented in Table 3.

All *P* values for the interaction term between GWG category, parental educational attainment level, and infant sex were >0.05 (see S7 Table). By comparing the baseline characteristics

**Table 2. Classification of weekly average gestational weight gain during the second and third trimesters and consistency among different guideline (n = 1,239).**

| Measure | GWG categories [n(%)] | | | Kappa | | |
|---|---|---|---|---|---|---|
| | **Below[a]** | **Within[a]** | **Above[a]** | | | |
| **IOM** | 798 (64.4) | 223 (18.0) | 218 (17.6) | Ref. | | |
| **NHC** | 579 (46.7) | 424 (34.2) | 236 (19.1) | 0.68 | Ref. | |
| **Z-score cat1[b]** | 804 (64.9) | 352 (28.4) | 83 (6.7) | 0.47 | 0.34 | Ref. |
| **Z-score cat2[c]** | 689 (55.6) | 493 (39.8) | 57 (4.6) | 0.45 | 0.43 | 0.78 |

Abbreviations: GWG, gestational weight gain; IOM, Institute of Medicine; NHC, National Health Commission.

[a]Below, within, and above classification were defined as below, within, and above the recommended range defined by IOM or NHC. For gestational–weight–gain–for–gestation–age z–scores derived from INTERGROWTH–21st Project, below, within, and above refer to subgroups 1,2,3, respectively.

[b]Z–score category 1 refers to subjects classified into three groups by percentiles of gestational–weight–gain–for–gestation–age z–score derived from INTERGROWTH–21st Project (z–score percentile<25th, 25th to 75th, >75th).

[c]Z–score category 2 refers to subjects classified into three groups by of gestational–weight–gain–for–gestation–age z–score derived from INTERGROWTH–21st Project (z–score<–1, –1 to 1, >1 SD).

**Table 3. Association between different GWG classifications and adverse birth outcomes among pregnant women with normal weight (n = 1,239).**

| | IOM category[a] | | NHC category[a] | | z-score category 1[ab] | | z-score category 2[ac] | |
|---|---|---|---|---|---|---|---|---|
| | Inadequate | Excessive | Inadequate | Excessive | z-score percentile<25th | z-score percentile>75th | z-score < -1 | z-score>1 |
| **Preterm birth** | 0.81 (0.35, 1.86) | 0.98 (0.35, 2.70) | 0.93 (0.46, 1.89) | 1.00 (0.41, 2.43) | 0.96 (0.48, 1.95) | 0.99 (0.26, 3.71) | 1.04 (0.54, 2.03) | 1.48 (0.40, 5.45) |
| **Post-term birth** | 2.09 (0.92, 4.72) | 1.77 (0.68, 4.64) | 2.25 (1.21, 4.16) | 1.64 (0.76, 3.55) | 0.93 (0.54, 1.60) | 0.42 (0.09, 1.82) | 1.23 (0.73, 2.07) | 0.36 (0.05, 2.76) |
| **LBW** | 1.25 (0.50, 3.09) | 1.08 (0.35, 3.35) | 1.03 (0.52, 2.02) | 0.80 (0.31, 2.04) | 1.10 (0.54, 2.27) | 1.52 (0.45, 5.11) | 1.12 (0.58, 2.17) | 1.49 (0.40, 5.55) |
| **Macrosomia** | 1.83 (0.39, 8.57) | 3.00 (0.55, 16.23) | 1.55 (0.44, 5.43) | 3.75 (1.03, 13.74) | 0.66 (0.23, 1.93) | 2.13 (0.48, 9.33) | 0.97 (0.35, 2.69) | 2.42 (0.45, 12.92) |
| **SGA** | 1.22 (0.75, 1.97) | 1.31 (0.73, 2.35) | 1.22 (0.83, 1.80) | 1.17 (0.72, 1.90) | 1.01 (0.68, 1.48) | 1.12 (0.55, 2.28) | 1.03 (0.72, 1.48) | 1.30 (0.59, 2.85) |
| **LGA** | 1.37 (0.58, 3.25) | 2.18 (0.84, 5.64) | 1.10 (0.55, 2.20) | 2.12 (1.01, 4.45) | 0.57 (0.31, 1.06) | 1.01 (0.37, 2.78) | 0.71 (0.39, 1.30) | 1.42 (0.48, 4.21) |

Abbreviations: GWG, gestational weight gain; IOM, Institute of Medicine; NHC, National Health Commission; LBW, low birth weight; SGA, small–for–gestational–age; LGA, large–for–gestational–age.

[a]Data are presented with adjusted odd ratios and 95% confidence intervals by performing generalized linear models. The adjustments included parental education, occupation and age, maternal parity, gestational week during early trimester when the maternal weight was measured, mid–upper arm circumference, randomized regimens and pre–pregnancy medical history, household wealth at enrollment, and infant sex.

[b]Z–score category 1 refers to subjects classified into three groups by percentiles of z–score (z–score percentile<25th, 25th to 75th, >75th). We set subgroup 2 (25th to 75th) as the reference group.

[c]Z–score category 2 refers to subjects classified into three groups by of z–score (<–1, –1 to 1, >1 SD). We set subgroup 2 (–1 to 1 SD) as the reference group.

between women included in the main analysis and those where not, we observed that women who were farmer, were more educated, had higher MUAC, or had higher household wealth were more likely to be included in the present analysis (see S8 Table). After applying IPW, the results revealed similar associations of inappropriate GWG defined by NHC and adverse birth outcomes (similar to Table 3 and S6 Table).

Furthermore, E-values for associations between inappropriate GWG and post-term birth, macrosomia, and LGA showed that our results were relatively robust given the unmeasured confounders e.g., depression, pregnant complication (see S9 Table).

## Discussion

This study found that half of the pregnant women with normal weight in rural western China gained gestational weight below the recommendation for either of the IOM or NHC guideline. The kappa value for the classification agreement between IOM and NHC guidelines was 0.68. After applying NHC guideline, inadequate GWG classified was positively associated with post-term birth, and excessive GWG was associated with higher risk of macrosomia and LGA compared with adequate GWG. When applied weight-gain-for-gestational-age z-score derived from the INTERGROWTH-21st Project, z-score was positively associated with birthweight while no significant association was observed between predefined subgroups of z-score and adverse birth outcomes.

### Recommended ranges for gestational weight gain

IOM guideline has been the most widely adopted for researches and clinical practices in many countries. As the increasing evidence examined the utility of IOM guideline across continents and ethnicities, especially in LIMCs settings [10, 11], several studies aimed to explore optimal

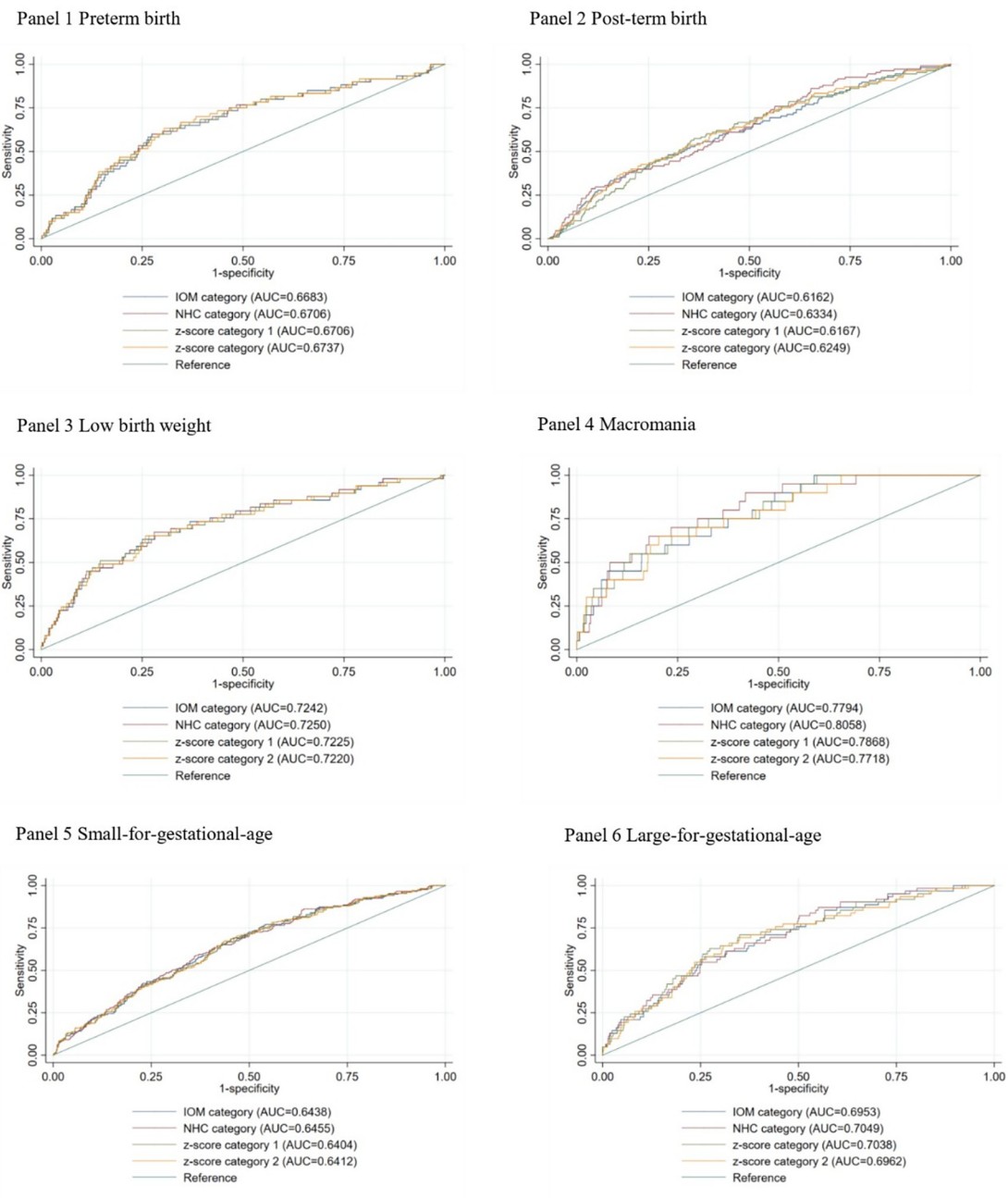

**Fig 1. The areas under the curve (AUC) for adverse birth outcomes predicted by each cut–off values of GWG.** Abbreviations: GWG, gestational weight gain; IOM, Institute of Medicine; NHC, National Health Commission. Z–score category 1 refers to subjects classified into three groups by percentiles of z–score (z–score percentile<25th, 25th to 75th, >75th). We set subgroup 2 (25th to 75th) as the reference group. Z–score category 2 refers to subjects classified into three groups by of z–score (<–1, –1 to 1, >1 SD). We set subgroup 2 (–1 to 1 SD) as reference group.

GWG ranges suitable for their regional population. In China, Zhang and colleagues recruited 3,731 women with singleton pregnancy from nine cities from 2013 to 2014 to establish optimal GWG ranges, which was defined as the range that did not exceed a 1% increase from the lowest predicted probability in each Chinese-specific pre-pregnancy BMI group [24]. Sun and colleagues enrolled 7,976 pregnant women from 24 hospitals in 15 provinces of China from 2017

to 2018 and then derived appropriate total GWG ranges for each Chinese-specific BMI group [14]. However, their samples did not include the population in rural western areas in China, and cut-offs proposed above have not been formally issued to guide pregnant women weight management clinically. In September 2021, the Chinese Nutrition Society (CNS) released the group standard *Weight Monitoring and Evaluation During Pregnancy Period of Chinese Women* (shown in S1 Table) [25]. And then, the NHC issued the health industry standard *Standard of Recommendation for Weight Gain during Pregnancy Period* in 2022, with the same cut-offs as CNS [15]. In the present study, the Kappa value of the GWG classification between the IOM and NHC recommendation was 0.68. More women with normal weight (defined as $18.5 \text{ kg/m}^2 \leq \text{BMI} < 24 \text{ kg/m}^2$ according to NHC guideline) were classified as inadequate GWG according to IOM than NHC, 64.4% versus 46.7%. For normal weight women, recommend GWG rate during the second and third trimesters in IOM guideline is slightly higher than that in NHC guideline (0.35–0.50 kg/week versus 0.26–0.48 kg/week). In addition, AUC values of adverse birth outcomes predicted by NHC cut-offs were numerical higher than those predicted by IOM guideline, although no statistically significant difference was observed. Therefore, when applying IOM recommendation to guide clinical weight management among Chinese pregnant women need more cautions. Further, we compared the consistency of categories classified by the NHC guideline and the INTERGRWTH-21st Project z-score according to two general approaches (z-score percentile<25th, 25th to 75th, >75th; z-score<-1, -1 to 1, >1 SD). The Kappa values for the classification agreement between NHC guideline and z-scores were only 0.43 and 0.34, which might due to the empirical cut-off approaches in the present study. Further studies examining the applicability and optimal ranges of z-scores derived from the INTERGROWTH-21st Project are in need.

## Gestational weight gain and birth outcomes

Previous studies have shown that the risk of post-term birth increase with increasing maternal BMI [26]. We overthrew the systematic review and meta-analyses focused on GWG and infant outcomes, while few pieces of evidence mentioned post-term birth [7, 27, 28]. In the present study, we found that inadequate GWG was associated with a higher risk of post-term birth among mothers with normal weight. Insufficient maternal energy intake may be linked causally with a greater risk of fetal growth restriction [29], and to enable the fetus achieve growth potential by prolonging the gestational week. Even in low-risk singleton pregnancies, post-term pregnancy appears to be an independent risk factor for adverse neonatal outcomes, including respiratory morbidity, infectious morbidity and hypoglycemia [30]. Additionally, it might have a lasting impact by raising the risk of developmental vulnerability through-out early childhood [31–33]. Therefore, post-term as well as preterm birth should also be considered as important neonatal outcomes and related risk factors should be explored in further studies.

A meta-analysis including 1,309,136 women revealed that GWG above the IOM recommendations showed a higher risk of LGA (OR 1.85; 95% CI 1.76, 1.95) and macrosomia (OR 1.95; 95% CI 1.79, 2.11) [7]. A study conducted in American found that higher GWG during the second and third trimesters was positively associated with the risk of LGA births among normal and underweight women, reflecting the period of in-utero weight and fat gain for fetus [34]. In the present study, we observed similar results. Maternal excessive GWG which tightly related to maternal high plasma concentrations of glucose, free fatty acids, and amino acids lead to permanent changes in energy metabolism in the developing fetus and thus lead to extremely high birth weight and even higher risk of obesity in later life [6, 35]. Taken together, these results emphasize the importance of weight management during the whole pregnancy

period, especially for the second and third trimesters, to avoid excessive GWG regardless of pre-pregnancy nutrition status.

The present study showed associations between GWG subgroups classified by NHC and adverse birth outcomes among women with normal weight. However, we did not observe corresponding associations of GWG classifications defined by IOM guideline and adverse birth outcomes, which might reflect the importance of establishing recommend ranges based on regional population. Zhang and colleagues proposed optimal total GWG ranges based on data from 3,731 women enrolled from nine cities in China and compared the proposed recommendations with the IOM guideline, and both of them indicated similar predictive values for adverse pregnancy outcomes [24]. While, cut-off values of their study were different from NHC released in 2022 (as shown in S1 Table). Chen and colleagues compared GWG guidelines released by the Chinese Nutrition Society (CNS), in which the recommended ranges are same as NHC, and IOM guideline by estimating the nonhigh nutritional status of the offspring using sensitivity, specificity, positive predictive value, and negative predictive value with respective to the mothers who kept appropriate GWG, and generally higher values based on CNS were found compared with those based on IOM recommendations among participants residing in Tianjin [36]. We enrolled participants from rural western areas of China, where suffered from disadvantaged socioeconomic status and the percentage of excessive GWG was <20%. Similarly, systematic review conducted among sub-Sahara Africa, where half of studies showed that the percentages of excessive GWG defined by IOM were <10%, and no significance was observed between excessive GWG and macrosomia [11]. In addition, our sample size was limited to pregnant women with normal weight during early pregnancy, which might reflect the need for caution in interpreting and comparing the results discussed above.

## Limitations

Several limitations should be noted. First, the mother-infant pairs in our cohort study derived from the prenatal cluster-randomized controlled trial, in which pregnant women received micronutrient supplementation (folic acid, IFA, and MMN), and consequently the generalization of the results might be limited. Recently, a meta-analysis of IPD among LMICs showed that maternal MMN supplementation was associated with greater GWG percentage adequacy of GWG (defined by IOM according to total GWG) than was iron and folic acid only [37]. Therefore, the intervention in our study might reduce the percentage of inappropriate GWG. We treated randomized regimens as a covariable in the models. Secondly, we limited analysis among those women enrolled less than 14 gestation weeks and without missing values of weight during the second or third trimesters which might lead to selection bias. We compared baseline characteristics of mother-infant pairs which enrolled in early pregnancy and those between more than 14 weeks' gestation (see S8 Table). Women who were farmer, were more educated, had higher MUAC, or had higher household wealth were more likely to be included in the present analysis. Thus, the proportion of inadequate GWG might be underestimate in the present study. In addition, we applied IPW to consider the potential of selection bias, and the results showed consistently association of inadequate GWG with post-term birth and excessive GWG with LGA baby. Thirdly, classifying GWG by maternal BMI during the first trimester instead of pre-pregnancy in the guidelines may result in misclassification to some extent. We adjusted the gestational age during early pregnancy when the maternal weight was measured in the models. Fourthly, unmeasured confounders may bias our results. For example pregnant complications e.g., gestational hypertension were associated with GWG by influencing diet patterns, physical activity, and other lifestyles [38, 39]. Although we failed to take pregnant complications as confounders into models in the present study, pre-pregnant disease

history including cardiac disease, kidney disease, chronic liver disease, hypertension, anemia and hyperthyroidism was considered. Furthermore, we applied E-values to assess unmeasured confounders, which showed acceptable robustness of our results. Finally, observational studies were disable to confirm the causality.

## Conclusions

Our findings suggest that more than half of pregnant women with normal weight were classified as having suboptimal gestational weight gain according to either of the GWG recommendation in rural western China. However, significant associations between suboptimal GWG and adverse birth outcomes were only observed according to the Chinese National Health Commission guideline promulgated in 2022. Monitoring the weight status of Chinese women during pregnancy under the proper range recommended by the Chinese National Health Commission were more suitable for population in rural western China, with potential benefits of improving birth outcomes. In addition, our findings suggest that establishing the local recommendations of appropriate GWG rather than applying the IOM is necessary for laying optimal health foundations for offspring life-course health. Furthermore, the optimal cut-off points of GWG z-scores derived from the INTERGROWTH-21$^{st}$ Project with their utility in global scale are urgently warranted in future studies.

## Supporting information

**S1 Table. Different recommendation ranges for gestational weight gains.**
(DOCX)

**S2 Table. Association between different GWG classifications and birth weight and gestational age among pregnant women with normal weight based on Chinese guidelines.**
(DOCX)

**S3 Table. AUC for gestational weight gain recommendation range corresponding to each adverse birth outcome, under IOM, NHC, and z-score, respectively.**
(DOCX)

**S4 Table. Association between different GWG and birth outcomes among pregnant women with normal weight based on Chinese guidelines.**
(DOCX)

**S5 Table. Classification of weekly average gestational weight gain during the second and third trimesters and consistency among different guidelines based on complete eligible data.**
(DOCX)

**S6 Table. Association between different GWG classifications and adverse birth outcomes among women with underweight, normal, overweight or obesity.**
(DOCX)

**S7 Table. Interaction P values between GWG categories and parental education and infant sex for adverse birth outcomes.**
(DOCX)

**S8 Table. Comparison of baseline characteristics between mother-infant pairs included into final analysis and those were not.**
(DOCX)

**S9 Table. Sensitivity analysis using inverse probability weight to assess the potential of selection bias of the association between different GWG classifications and adverse birth outcomes among women with underweight, normal, overweight or obesity.**
(DOCX)

**S10 Table. E-values for the association between different GWG classifications and adverse birth outcomes among pregnant women with normal weight.**
(DOCX)

**S1 Fig. Flowchart.**
(TIF)

## Acknowledgments

We thank all field workers who helped with data collection. We are also grateful to all participants and their families. The authors thank AiMi Academic Services (www.aimieditor.com) for English language editing and review services.

## Author Contributions

**Conceptualization:** Yue Cheng, Zhonghai Zhu, Lingxia Zeng.

**Data curation:** Liang Wang, Qi Qi, Lingxia Zeng.

**Formal analysis:** Yingze Zhu.

**Investigation:** Liang Wang, Qi Qi, Zhonghai Zhu.

**Methodology:** Yue Cheng, Zhonghai Zhu, Lingxia Zeng.

**Supervision:** Lingxia Zeng.

**Writing – original draft:** Yingze Zhu.

**Writing – review & editing:** Zhonghai Zhu.

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
