## [Decision Letter · Decision Letter 0]

13 Jul 2023

PGPH-D-23-00983

Associations between gestational weight gain in different guidelines and adverse birth outcomes: a secondary analysis of a randomized controlled trial in rural western China

Dear Dr. Zeng,

Thank you for submitting your manuscript to PLOS Global Public Health. After careful consideration, we feel that it has merit but does not fully meet PLOS Global Public Health’s publication criteria as it currently stands. Therefore, we invite you to submit a revised version of the manuscript that addresses the points raised during the review process.

Please note that we have only been able to secure a single reviewer to assess your manuscript. We are issuing a decision on your manuscript at this point to prevent further delays in the evaluation of your manuscript. Please be aware that the editor who handles your revised manuscript might find it necessary to invite additional reviewers to assess this work once the revised manuscript is submitted. However, we will aim to proceed on the basis of this single review if possible. 

Could you please revise the manuscript to carefully address the concerns raised? (See comments below and in the attached file).

We look forward to receiving your revised manuscript.

Kind regards,

Steve Zimmerman, PhD

PLOS Staff Editor

Journal Requirements:

Additional Editor Comments (if provided):

Reviewers' comments:

Reviewer's Responses to Questions

**Comments to the Author**

1. Does this manuscript meet PLOS Global Public Health’s publication criteria? Is the manuscript technically sound, and do the data support the conclusions? The manuscript must describe methodologically and ethically rigorous research with conclusions that are appropriately drawn based on the data presented.

Reviewer #1: Yes

2. Has the statistical analysis been performed appropriately and rigorously?

Reviewer #1: Yes

3. Have the authors made all data underlying the findings in their manuscript fully available (please refer to the Data Availability Statement at the start of the manuscript PDF file)?

Reviewer #1: Yes

4. Is the manuscript presented in an intelligible fashion and written in standard English?

Reviewer #1: Yes

5. Review Comments to the Author

Reviewer #1: This study found a big data to discuss about the different gestational weight gain recommendation in the western chinese women population. the length of discussion need to be divided into several sub-topics. the presentation data need to be improved with some of data present in figure. all comments are attached in the pdf file.

6. PLOS authors have the option to publish the peer review history of their article (what does this mean?). If published, this will include your full peer review and any attached files.

**Do you want your identity to be public for this peer review?** For information about this choice, including consent withdrawal, please see our Privacy Policy.

Reviewer #1: **Yes: **Arif Sabta Aji

---

## [Decision Letter · Decision Letter 1]

16 Oct 2023

PGPH-D-23-00983R1

Associations between gestational weight gain in different guidelines and adverse birth outcomes: a secondary analysis of a randomized controlled trial in rural western China

Dear Dr. Zeng,

Thank you for submitting your manuscript to PLOS Global Public Health. After careful consideration, we feel that it has merit but does not fully meet PLOS Global Public Health’s publication criteria as it currently stands. Therefore, we invite you to submit a revised version of the manuscript that addresses the points raised during the review process.

Reviewer #2: 1. There is a lack of rationale explaining why different guidelines are associated with adverse birth outcomes. The rationale together with specific hypotheses, including why you are comparing two guidelines, should be clearly stated in the Introduction.

2. How covariates were chosen should be described in more detail in the methods section. However, the operationalization of these variables should be provided in the text.

3. The analysis was based on data from a longitudinal cohort study. What is the attrition rate? Attrition analysis should be conducted and discussed.

4. Also clearly mention about co-morbid condition such as cardiac disease, hypertension, diabetes, tuberculosis, kidney disease, chronic 201 liver disease, hyperthyroidism and or anemia in line number 200-202.

5. In discussion, line number 320-325 it’s a repetition of result and second para, line number 329-325 its not link with whole discussion. Need more justification why we need this study.

We look forward to receiving your revised manuscript.

Kind regards,

Nazmul Alam, MPH, DrPH

Academic Editor

---

## [Editor Report · Decision Letter 2]

14 Nov 2023

Associations between gestational weight gain in different guidelines and adverse birth outcomes: a secondary analysis of a randomized controlled trial in rural western China

PGPH-D-23-00983R2

Dear Dr. Zeng,

We are pleased to inform you that your manuscript 'Associations between gestational weight gain in different guidelines and adverse birth outcomes: a secondary analysis of a randomized controlled trial in rural western China' has been provisionally accepted for publication in PLOS Global Public Health.

Best regards,

Nazmul Alam, MPH, DrPH

Academic Editor